# Impact of Psychotherapy on Psychosocial Functioning in Borderline Personality Disorder Patients

**DOI:** 10.3390/ijerph17124610

**Published:** 2020-06-26

**Authors:** Soheil Zahediabghari, Philippe Boursiquot, Paul Links

**Affiliations:** Department of Psychiatry and Behavioural Neurosciences, McMaster University, Hamilton, ON L8S 4L8, Canada; zaheds1@mcmaster.ca (S.Z.); pboursiqu@stjosham.on.ca (P.B.)

**Keywords:** Borderline Personality Disorder (BPD), psychotherapy, psychosocial functioning

## Abstract

Borderline Personality Disorder (BPD) significantly impairs functioning. Fortunately, effective treatments are available for borderline symptoms but their effect on functioning should be assessed. The objective of this meta-analysis is to assess the effect of specifically-designed versus non-specifically designed psychotherapies on function in adult patients with BPD. The reference list of Cristea et al. 2017 was used to identify the randomized controlled trials (RCTs) assessing the BPD-specifically-designed psychotherapy versus non-specific psychotherapies in adult BPD patients. Among those, RCTs assessing post-treatment functioning using the Global Assessment of Functioning, Social Adjustment Scale–Self-Report and Inventory of Interpersonal Problems were included. Ten trials (880 participants) were included. Summary effect size was calculated using the measured Hedge’s g. The results indicate the BPD patients in the intervention group had a significantly higher (g = 0.41; 95% CI, 0.09–0.73) level of psychosocial functioning after receiving the specifically-designed psychotherapies in comparison with BPD patients in control groups after receiving non-specific psychotherapies. Specifically-designed psychotherapies can improve psychosocial functioning although improvement in measurement of function (i.e., more objective and universal tools) and improvement in psychotherapies (i.e., more focused on general functioning) will be helpful.

## 1. Introduction

Borderline Personality Disorder (BPD) significantly impairs psychosocial functioning and this is regardless of individuals’ gender [1] or settings (general population or clinical/hospital settings) [2]. By nature, BPD disturbs both self and interpersonal domains causing severe reduction in the level of psychosocial functioning [3]. This degree of impairment is significantly greater than that of depression when assessed in the clinical population [4]. Furthermore, when considering the clinical course of BPD over a ten-year period, psychosocial functioning is severely impaired with long-lasting effects while symptoms of BPD more readily remit and infrequently relapse [5].

BPD is common and almost 6 people out of 100 will have BPD in the course of their lives [6]. Approximately 1% of people in society have BPD at any given time [7]. The proportion of BPD patients increases to 10–22% when it comes to individuals seeking help in mental health clinics and hospitals [7].

Effective treatments exist to improve BPD related problems and symptoms [8] but the effect of BPD treatment on psychosocial functioning has yet to be determined [9]. Many psychotherapies have been developed particularly to assist patients dealing with BPD. These specifically designed psychotherapies including dialectical behavior therapy (DBT), cognitive behavior therapy (CBT) and psychodynamic approaches, like mentalization-based therapy (MBT) and transference-focused psychotherapy(TFP), are shown to improve BPD symptoms [8]. Unfortunately, psychosocial function in BPD is often insufficiently prioritized in clinical practice, treatment and research [2]. Research is needed to assess the effect of BPD treatment on psychosocial functioning [9]. Furthermore, long-term follow up studies indicated that poor psychosocial functioning continues [10] emphasizing the importance of assessing the impact of BPD treatment on psychosocial functioning. The objective of this review is to assess the effect of psychotherapies specifically designed for BPD on function in adult patients with BPD.

## 2. Methods

### 2.1. Identification and Selection of Studies

The list of included studies in the Cristea et al. (2017) [8] meta-analysis was used to identify RCTs (Randomized Controlled Trials) to include in our study. No other RCTs in adults have been published since Cristea et al. up to 16 June 2020 that would meet the inclusion criteria of this review. These RCTs assessed psychotherapies specifically developed for BPD (mostly DBT, CBT and psychodynamic approaches) versus non-specifically-designed psychotherapies (mainly treatment as usual and supportive treatment) in adult BPD patients. Among these studies, the ones which had all of the following three criteria were included: (i) assessed psychosocial functioning at post-treatment timepoint; (ii) used at least one of these three tools to assess psychosocial functioning: Global Assessment of Functioning (GAF), Social Adjustment Scale–Self-Report (SAS-SR) and Inventory of Interpersonal Problems (IIP) (we have chosen these tools because these were the only tools that were used in more than one RCTs to assess psychosocial functioning); (iii) the mean score and either the Standard Deviation (SD) or Standard Error of Mean (SEM) were available, either in the main published article or in the article supplement that they have released.

Cristea et al. (2017) used borderline personality keywords to search for randomized control trials in PubMed, PsycINFO, EMBASE, and the Cochrane Central Register (from the date of creation of the database to November 2015). They included RCTs which compared specifically-designed psychotherapies for BPD versus non-specifically-designed psychotherapies in adult populations. This inclusion was regardless of participants’ status of using medication.

### 2.2. Risk of Bias and Data Extraction

Cristea et al. (2017) [8] assessed bias in RCTs using Trial Risk of Bias (RoB) (which is presented in the eFigure 2 of supplement content of their article [8]). They also have reported ratings of bias for each of their included studies which are available in their article supplement. Considering RCTs included in our study, the Rating of Bias ranges from 0 in Bateman and Fonagy (1999) [11] to 4 in Bateman and Fonagy (2009) [12] and McMain et al. (2009) [8,13]. Farrell et al. (2009) [14] and Jorgensen et al. (2013) [15] had 1 as their rating of bias while Amianto et al. (2011) [16] and Blum et al. (2008) [17] had 2 [8]. The three remaining studies, Davidson et al. (2006) [18], Doering et al. (2010) [19] and Kramer et al. (2014) [20] were rated 3 [8].

Typically, publication bias is assessed using funnel plot techniques, Begg’s rank test and Egger’s regression test (see page 5 for a discussion of publication bias in the current study).

### 2.3. Meta-Analysis

The study articles were used to extract post-treatment outcome data. These data include the number of treated participants, the mean score results of psychosocial measurement tools, and Standard Deviation (SD) or Standard Error of the Mean (SEM). Comprehensive Meta-Analysis Software (CMA) was used to calculate the effect size and Hedge’s g. In order to calculate the effect size, we used the random-effect model. The same software, CMA, was also used to create forest plots as well as calculate heterogeneity.

## 3. Results

### 3.1. Selection and Inclusion

The full text articles of 33 RCTs which were previously included in Cristea et al. (2017) [8] were examined for inclusion in this study. Among those, ten trials with 880 intention-to-treat participants were included. 23 studies were excluded for the following reasons: 17 did not measure post-treatment psychosocial functioning, 3 studies used SCL-90-R which assesses more symptoms rather than psychosocial functioning, 1 used the Functioning Assessment Short Test (FAST) and this study was excluded because it was the only RCT using this tool, and 2 did not have enough information in the published article. The selection of studies and data extraction were reviewed by all investigators to ensure consensus.

### 3.2. Characteristics of Included Studies

In the ten included RCTs, 880 intention-to-treat participants were recruited from adult individuals who were diagnosed with Borderline Personality Disorder (BPD) Structured Clinical Interview for DSM (SCID) except for Farrell et al. (2009) who used Diagnostic Interview for Borderline Personality Disorders-Revised (DIBP-R) and Borderline Syndrome Index (BSI). Recruitment took place mostly in outpatient and inpatient psychiatric units as well as community mental health care settings.

Treatments in intervention groups are demonstrated in Table 1. In 5 RCTs, treatment in intervention groups were stand-alone, i.e., the participants received only the specifically designed treatment for BPD and in 5 RCTs (Table 1), the intervention group received both specifically-designed psychotherapy plus the treatment of the control group. Treatments in control groups were mainly treatment as usual and supportive treatment (Table 1).

The outcome measurement tools used are Global Assessment of Functioning (GAF), Social Adjustment Scale-Self-Report (SAS-SR), and Inventory of Interpersonal Problems (IIP) shown in Table 1. The higher the GAF score, the higher the level of functioning while the lower the IIP or SAS-SR score the higher the level of functioning.

### 3.3. Findings

The results indicate the BPD patients in the intervention group had a higher (g = 0.41; 95% CI, 0.09–0.73) level of psychosocial functioning after receiving the specifically-designed psychotherapies in comparison with BPD patients in control groups who received non-specifically-designed psychotherapies. Hedge’s g for the individual studies is presented in Figure 1.

In calculation of heterogeneity, there is much uncertainty when a small number of studies are included in a meta-analysis, like ours. Although there is no specific number of studies as the cutoff point in this regard, it seems that I-squared is less affected by the small number of studies included although it should be considered that I-squared is rather a more descriptive statistic than point estimate [21]. In our study, I-squared was high (I2 = 80%) which shows a substantial heterogeneity. A part of this heterogeneity can be explained by the variability of measurement tools, especially when comparing the heterogeneity to that of Cristea et al. (2017) [8], where post-test BPD relevant outcomes (BPD symptoms plus suicidality and para-suicidality) heterogeneity is I2 = 48%. In this comparison, the number of studies used in our meta-analysis should also be considered, which is 10 studies, a subset of the 27 studies in Cristea et al. (2017).

In our study, funnel plot or more advanced regression-based assessments were not calculated to examine the publication bias and the reason is that the number of trials included in our study was limited which could cause unreliable results.

## 4. Discussion

The results of this meta-analysis show psychotherapies specifically designed for BPD can improve psychosocial functioning. The results indicate the BPD patients in the intervention group had a significantly higher (g = 0.41; 95% CI 0.09–0.73) level of psychosocial functioning after receiving the specifically-designed psychotherapies in comparison with BPD patients in control groups after receiving non-specific psychotherapies. Specifically-designed psychotherapies can improve psychosocial functioning although improvement in measurement of function (i.e., more objective and universal tools) and improvement in psychotherapies (i.e., more focused on general functioning) will be helpful in future intervention studies. The importance of this improvement will be even more apparent when considering that long term follow up studies indicated poor functioning continues [10] while symptoms more readily remit and infrequently relapse [5]. Although limited to ten studies, this review may mean that the specifically-designed psychotherapies play an important role not only in the symptoms of BPD patients but their function as well. However, the wider confidence interval in our study related to functioning rather than symptom outcomes should be considered. Generally, the width of confidence interval in a meta-analysis with random-effects model depends on the number of studies combined, the heterogeneity in the meta-analysis and variability in the outcome measurements [22]. The wide confidence interval (CI) in our meta-analysis indicates the need for more studies. Furthermore, using more universal psychosocial function measurements not only lowers the level of heterogeneity but also makes it possible to include more studies in future meta-analyses which, in turn, helps to compare interventions more effectively.

It is important to point out a few limitations of the research: (I) considerable variability is seen in the results of the functioning assessments; (II) many different measurement tools are used to assess functioning and more universally accepted tools are needed; for example GAF was the most commonly used measure along with the Inventory of Interpersonal Problems (6/10 studies utilized this measure to assess psychosocial functioning); however, the GAF combines both symptoms and function and has been criticised as measure of function [23]; (III) measurements tools are self-reported which creates the need for more objective tools and multiple methods of measurement of function; (IV) risk of bias in outcome level and review levels should be examined; (V) limited number of RCTs are included which has caused the wide confidence interval so more studies are needed; (VI) high degree of heterogeneity exists although there is much doubt on the interpreting heterogeneity statistic when the number of studies included is low, like our study; (VII) potential publication bias should be considered which is not calculated due to inaccuracy of available methods when the number of included RCTs is low.

## 5. Conclusions

The meta-analysis result shows a higher level of functioning in the treatment group. This can challenge the impression that BPD psychotherapies may not impact patients’ functioning.

Although the results show the specific therapies lead to improvement in functioning, the following points about tools and interventions should be considered. Firstly, considerable variability is seen when the results of the three different functioning assessment tools are compared. This indicates the potential need for an agreed upon measurement tool. A more universal tool will make it easier to compare different studies and also will create a more common language among clinicians and researchers. Developing such a consensus will make it more practical to consider psychosocial functioning in clinical assessments, treatment and research. Secondly, the current measurement tools are self-reported which can be a source of bias indicating the need for more objective and multiple methods of assessment.

Lastly, interventions are mostly focused on BPD symptoms. Until further empirical evidence is established, the study may suggest approaches to clinical care for patients with BPD. Clinicians are well advised to have their patients focus on life outside of treatment and particularly to develop goals that foster productive vocational activities. When medications are prescribed, clinicians should look for evidence that the psychopharmacological intervention has an impact on measurable functional improvement. Interventions should focus more on work function and function in spousal and interpersonal relations to improve global functioning in BPD patients, as well as the barriers (psychological and social, e.g., sufficient income) to improving functioning. Additionally, the interplay between such barriers and BPD symptoms also merits further research.

To that end, Level of Personality Functioning Scale (LPFS) is a promising psychosocial functioning measurement method which evaluates the global level of impairment in personality functioning. Several measures of function have been developed and validated based on the alternative DSM-5 model for personality disorders that hopefully will be useful in clinical trials [24].

Stepped care is a useful and known method of treatment in mental health disorders which is particularly useful for BPD. Adding steps in BPD treatments to specifically focus on spousal and interpersonal relations and work functioning could help to enhance the beneficial impacts of BPD treatments.

## Figures and Tables

**Figure 1 ijerph-17-04610-f001:**
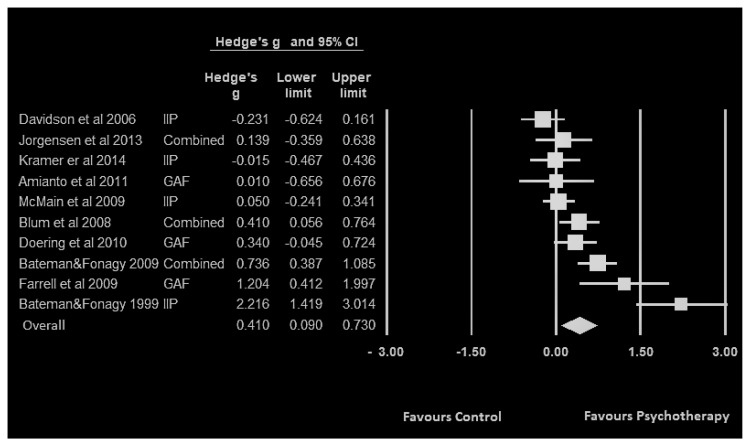
Forest plot and Hedge’s g (effect size) of each included study with their 95% confidence Interval (CI) upper limit and lower limit comparing psychosocial functioning of the BPD specific treatments in the intervention group versus non-specific treatment in the control group in the ten included studies.

**Table 1 ijerph-17-04610-t001:** Types of psychotherapies used in intervention and control groups in the included studies, along with outcome measurements used in the meta-analysis.

Included Studies	Tools *	Trt in Intervention Grp+	Trt in Ctrl Grp++
Amianto et al. 2011 [16]	GAF	SB-APP + STM	STM
Bateman and Fonagy 1999 [11]	IIP	MBT-PH	SPT
Bateman and Fonagy 2009 [12]	GAF, IIP, SAS-SR	MBT	SCM
Blum et al. 2008 [17]	GAF, SAS-SR	STEPPS + TAU	TAU
Davidson et al. 2006 [18]	IIP	CBT + TAU	TAU
Doering et al. 2010 [19]	GAF	TFP	ECP
Farrell et al. 2009 [14]	GAF	SFT + TAU	TAU
Jorgensen et al. 2013 [15]	GAF, SAS-SR, IIP	MBT	ST
Kramer et al. 2014 [20]	IIP	MOTR + GPM	GPM
McMain et al. 2009 [13]	IIP	DBT	GPM

* These are psychosocial function assessment tools the results of which were available in the published articles. +Psychotherapies used in the intervention group, ++Psychotherapies used in the control group. Abbreviation: Trt (Treatment), Grp (Group), Ctrl (Control), GAF (Global Assessment of Functioning), SAS-SR (Social Adjustment Scale–Self-Report), IIP (Inventory of Interpersonal Problems), SB-APP (Sequential Brief Adlerian Psychodynamic Psychotherapy), MBT (Mentalization-based treatment), PH (partial hospitalization), STEPPS (Systems Training for Emotional Predictability and Problem Solving), CBT (Cognitive Behavior Therapy), TFP (Transference Focused Psychotherapy), MBT (Mentalization-based treatment), MOTR (Motive-Oriented Therapeutic Relationship), DBT (Dialectical Behaviour Therapy), STM (Supervised Team Management), SPT (Standard Psychiatric Care), SCM (Structured Clinical Management), TAU (Treatment As Usual), ECP (Experienced Community Psychotherapists), ST (Supportive Therapy), GPM (General Psychiatric Management).

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
