# Peer review of "Impact of Psychotherapy on Psychosocial Functioning in Borderline Personality Disorder Patients"

_ijerph, 2020, doi:10.3390/ijerph17124610_

Round 1

Reviewer 1 Report

This paper is a summarized paper which focuses on ten trials and 880 participants’ treatment functioning evaluation based on meta-analysis. Of course, meta-analysis is a normal evaluation method to discuss the bias among different published papers and reports. However, some of the psychology measurement tools such as Scale-Self-Report tend to subjective judgment at the most, but the influence of objective factors are the main causes for BPD diagnosis. I mean we need more intelligent methods to deeply analysis the risk factors’ correlation based on more dimensions and to judge whether the patient actually suffers from BPD, not only discussing the bias with different measurement methods.

In fact, the dataset is from 1999 to 2014, which presents bias is not only causes by measurement and evaluation methods, but also causes by different history periods’ social factors. It means different periods of social development leads different kinds of BPD patients, which cause by different risk social factors. Moreover, if the dataset is come from recent 5 or 10 years, it looks more convincing and helpful.

The result is so simplistic that it seems obvious. The figure 1 is not very clear.

Author Response

June 17, 2020

Manuscript ID: ijerph-837437
Type of manuscript: Review
Title: Impact of Psychotherapy on Psychosocial Functioning in Borderline 
Personality Disorder Patients
Authors: Soheil Zahediabghari, Philippe Boursiquot, Paul Links *
Received: 31 May 2020

Dear Editor:

We very much appreciate the thoughtful reviews received on our paper and we have addressed reviewer 1’s comments as outlined in this letter of response: 

“However, some of the psychology measurement tools such as Scale-Self-Report tend to subjective judgment at the most, but the influence of objective factors are the main causes for BPD diagnosis. I mean we need more intelligent methods to deeply analysis the risk factors’ correlation based on more dimensions and to judge whether the patient actually suffers from BPD, not only discussing the bias with different measurement methods.”

In the discussion we mention “measurements tools are self-reported which

creates the need for more objective tools” and we have added “and multiple methods of measurement of function.”

“if the dataset is come from recent 5 or 10 years, it looks more convincing and helpful.”

We have updated our search for other studies that would meet the inclusion criteria of our review and can confirm that as of June 16, 2020, no other RCTs in adults have been published since Cristea et. al paper. We have identified a RCT study protocol by Hurtado-Santiago et al. (2018) examining Iconic Therapy compared to Structured Support Therapy in youth (ages 15-25) with BPD symptoms, as well as a RCT by Beck et al. (2020) comparing the effect of Mentalization-based Treatment in adolescents with BPD.

Reviewer 2 Report

Manuscript ID: ijerph-837437

I am grateful for the opportunity of analyzing your work about impact of psychotherapy on psychosocial functioning in patients with BPD. This current manuscript underlines as a main result that specifically-designed psychotherapies have better results in the level of psychosocial functioning compared with those patients receiving non-specific psychotherapies.

Current study has interesting importance and relevant clinical use. However, there some issues that would have to be considered for its improving.

Minor details.

Inside title, meta-analysis word is not reflected as something that is consistently marked as a methodological strategy inside the text.

Although introduction is concise, maybe it is needed an update on this topic in last two years.

On methodology section, statistic strategy used is clear.

In results, Figure 1 is not mentioned inside text.

Figure 1 seems a replica of the one mentioned by Cristea et al.2017 e Figure2, coming from studies included in its qualitative synthesis (n=33).

In section 3.3 it would be recommendable to present figure2 in black/white format.

Conclusions are interesting and well cohesive with the whole of the manuscript.

Has manuscript appendices?, A, A,1 and B are mentioned, but I do not identify them clearly.

Global listing of references needs a greater update, but it only has 5 publications out of a 23 total in the last three years.

Major details.

I think that besides suggested limitations, one of the greater magnitude is basing this current study of meta-analysis performed in 2020 in results only found by Cristea et al. in 2017. Having into account that the study of Cristea et al. only includes studies' analysis up to November 2015. Therefore, in last years (almost 5 years; at least from 2016 to 2019) perhaps there has been  some advance in this area, I think that authors would have to justify better the limitation of this cut in date, for potentially information obtained  based on this meta-analysis could be not updated.

In the other hand, in discussion section they compare sizes of psychosocial functioning effect (g=0.41;95% CI, 0.09-0.73 with RCTs) with the ones from Cristea et al. 2017(g=0.35; 95%CI, 0.20-0.50),  authors comment several possible interpretations. However, they do not comment that within their selection (based on the criteria of this present study) they exclude 2 studies with the worst previous results found in Cristea et.al. 2017 (specifically Cottraux et al. 2009, g=-0.48; Pascual et al. 2015, g=-0.73) therefore I do not know if comparison is effective by itself, when assessing Borderline-Relevant Outcomes. Even though they mark within discussion that is necessary to perform broader studies in the future, precisely this is what would be interesting to develop in this current manuscript (including different studies more updated).

Author Response

June 17, 2020

Manuscript ID: ijerph-837437
Type of manuscript: Review
Title: Impact of Psychotherapy on Psychosocial Functioning in Borderline 
Personality Disorder Patients
Authors: Soheil Zahediabghari, Philippe Boursiquot, Paul Links *
Received: 31 May 2020

Dear Editor:

We very much appreciate the thoughtful reviews received on our paper and we have addressed reviewer 2's comments as outlined in this letter of response:

Inside title, meta-analysis word is not reflected as something that is consistently marked as a methodological strategy inside the text.

We have used meta-analysis wording throughout the document.

Although introduction is concise, maybe it is needed an update on this topic in last two years.

We have updated our search for other studies that would meet the inclusion criteria of our review and can confirm that as of June 16, 2020, no other RCTs in adults have been published since Cristea et. al paper. We have identified a RCT study protocol by Hurtado-Santiago et al. (2018) examining Iconic Therapy compared to Structured Support Therapy in youth (ages 15-25) with BPD symptoms, as well as a RCT by Beck et al. (2020) comparing the effect of Mentalization-based Treatment in adolescents with BPD.

On methodology section, statistic strategy used is clear.

In results, Figure 1 is not mentioned inside text.

Figure 1 seems a replica of the one mentioned by Cristea et al.2017 e Figure2, coming from studies included in its qualitative synthesis (n=33).

Figure 1 is not a duplication of Cristea et al. flow chart of selection and inclusion but demonstrates how we selected the appropriate studies from the original study leading to our inclusion of 10 studies.

In section 3.3 it would be recommendable to present figure2 in black/white format.

We have revised Figure 2 to use black/white format.

Conclusions are interesting and well cohesive with the whole of the manuscript.

Has manuscript appendices?, A, A,1 and B are mentioned, but I do not identify them clearly.

No appendices have been submitted with our paper.

Global listing of references needs a greater update, but it only has 5 publications out of a 23 total in the last three years. “Therefore, in last years (almost 5 years; at least from 2016 to 2019) perhaps there has been  some advance in this area, I think that authors would have to justify better the limitation of this cut in date, for potentially information obtained  based on this meta-analysis could be not updated.”

We have added two additional references e.g. Monrad Aas, 2010; Waugh et al. 2020 but as indicated above there are no more recent trials to include.

“I do not know if comparison is effective by itself, when assessing Borderline-Relevant Outcomes”

We agree that this comparison is misleading because the studies considered in the two meta-analyses are different and we have removed this comment.

“Even though they mark within discussion that is necessary to perform broader studies in the future, precisely this is what would be interesting to develop in this current manuscript (including different studies more updated).”

As discussed above, we have updated our search and can confirm that no other RCTs in adults have been published since Cristea et. al paper.

Reviewer 3 Report

See attached file for detailed comments.

Author Response

June 17, 2020

Manuscript ID: ijerph-837437
Type of manuscript: Review
Title: Impact of Psychotherapy on Psychosocial Functioning in Borderline 
Personality Disorder Patients
Authors: Soheil Zahediabghari, Philippe Boursiquot, Paul Links *
Received: 31 May 2020

Dear Editor:

We very much appreciate the thoughtful reviews received on our paper and we have addressed reviewer 3’s comments as outlined in this letter of response: 

- “main reservation relates to the inclusion of studies that have exclusive used the GAF scale”- We agree with the reviewer’s criticism of the GAF scale and we have added this issue to the limitations section that the GAF measures both symptoms and function and added the reference provided by the reviewer (Monrad Aas, 2010). However, we have chosen to continue to include GAF studies as it was the most commonly used measure along with the Inventory of Interpersonal Problems (6/10 studies utilized this measure to assess psychosocial functioning) and highlights the need for utilization of better measures in future research.

-they mention that “Specifically-designed psychotherapies, contrary to common belief, …” We have removed the phrase "contrary to common belief" as we agree that this was overstating the point. Some randomized trials have failed to demonstrate change in psychosocial outcomes (e.g. McMain et al. 2008) but psychotherapy interventions are not ignoring psychosocial elements as the reviewer correctly points out.

- I do not believe that there have been any major randomized studies with the potential for inclusion…”

We have updated our search for other studies that would meet the inclusion criteria of our review and can confirm that as of June 16, 2020, no other RCTs in adults have been published since Cristea et. al paper. We have identified a RCT study protocol by Hurtado-Santiago et al. (2018) examining Iconic Therapy compared to Structured Support Therapy in youth (ages 15-25) with BPD symptoms, as well as a RCT by Beck et al. (2020) comparing the effect of Mentalization-based Treatment in adolescents with BPD.

“Some confusion surrounds the publication bias”

We have clarified our comments in the following way “In other studies, publication bias is assessed using funnel plot techniques, Begg’s rank test and Egger’s regression test (see page 5 for a discussion of publication bias in the current study)”. Under the “Findings” on page 5, we comment on our approach. 

…offers a more or less convincing comparison with the results of the meta-analysis by Cristea et al. 

We agree that this comparison is misleading because the studies considered in the two meta-analyses are different and we have removed this comment.

Some clinical reflections … may be in order

We have added to the discussion “Until further empirical evidence is established, clinicians are well advised to have their patients focus on life outside of treatment and particularly to develop goals that foster productive vocational activities. When medications are prescribed, clinicians should look for evidence that the psychopharmacological intervention has an impact on measurable functional improvement.”

More exactly several measures have already been developed and validated… 

We have revised our statement and added the suggested reference (Waugh et al. 2020) “Several measures of function  have been developed and validated based on the alternative DSM-5 model for personality disorders that hopefully will be useful in clinical trials.” 

Punctuation would benefit from being reviewed … and standardized Hedge’s g

We have reviewed the punctuation and have used “Hedge's g” throughout the paper.

Round 2

Reviewer 1 Report

There without many substantive revisions for the previous reviews, even for the unclear figure 1.

The results of meta-analysis and discussion are not very persuasive.

Author Response

June 21, 2020

Manuscript ID: ijerph-837437
Type of manuscript: Review
Title: Impact of Psychotherapy on Psychosocial Functioning in Borderline 
Personality Disorder Patients
Authors: Soheil Zahediabghari, Philippe Boursiquot, Paul Links *
Received: 31 May 2020

Dear Editor:

We very much appreciate the thoughtful reviews received on our paper and we have addressed the second round of reviewer 1’s comments in this letter of response: 

- “There without many substantive revisions for the previous reviews, even for the unclear figure 1.”- In the second round, we removed Figure 1 and described the process in the text.

- “The results of meta-analysis and discussion are not very persuasive.”- The other two reviewers considered the work as important, relevant and clinically useful.

Thank you for your time and assistance. 

Sincerely,

Dr. Paul Links, MD, FRCPC
